# Role of the Myokine Irisin on Bone Homeostasis: Review of the Current Evidence

**DOI:** 10.3390/ijms22179136

**Published:** 2021-08-24

**Authors:** Amanda Kornel, Danja J. Den Hartogh, Panagiota Klentrou, Evangelia Tsiani

**Affiliations:** 1Department of Health Sciences, Faculty of Applied Health Sciences, Brock University, St. Catharines, ON L2S 3A1, Canada; ak12af@brocku.ca (A.K.); dd11qv@brocku.ca (D.J.D.H.); 2Centre for Bone and Muscle Health, Brock University, St. Catharines, ON L2S 3A1, Canada; nklentrou@brocku.ca; 3Department of Kinesiology, Brock University, St. Catharines, ON L2S 3A1, Canada

**Keywords:** myokines, irisin, bone, osteoblasts, osteoclasts, osteocytes, osteoporosis, in vitro, in vivo, clinical

## Abstract

Bone is a highly dynamic tissue that is constantly adapting to micro-changes to facilitate movement. When the balance between bone building and resorption shifts more towards bone resorption, the result is reduced bone density and mineralization, as seen in osteoporosis or osteopenia. Current treatment strategies aimed to improve bone homeostasis and turnover are lacking in efficacy, resulting in the search for new preventative and nutraceutical treatment options. The myokine irisin, since its discovery in 2012, has been shown to play an important role in many tissues including muscle, adipose, and bone. Evidence indicate that irisin is associated with increased bone formation and decreased bone resorption, leading to reduced risk of osteoporosis in post-menopausal women. In addition, low serum irisin levels have been found in individuals with osteoporosis and osteopenia. Irisin targets key signaling proteins, promoting osteoblastogenesis and reducing osteoclastogenesis. The present review summarizes the existing evidence regarding the effects of irisin on bone homeostasis.

## 1. Introduction

### 1.1. Bone Homeostasis

Bone is a highly dynamic tissue that requires it to be stiff yet flexible, and light yet strong in order to facilitate movement and withstand loading. Bone also has a specific characteristic that separates it from other tissues, in that it can respond to damage according to its location and magnitude, by removing old tissue and creating new bone in its place [1]. Bone is comprised of cells (osteoblasts, osteoclasts, and osteocytes), and an extracellular matrix which is mineralized by calcium hydroxyapatite deposition to give bone its rigidity and strength [2,3]. Osteoblasts, also known as the bone-forming cells, are derived from mesenchymal stem cells and express specific genes, such as bone morphogenetic proteins (BMPs) and members of the Wingless (Wnt) pathway [4]. Differentiated osteoblasts secrete factors that regulate the differentiation of osteoclasts, also known as the bone-resorbing cells, influencing the balance between bone formation and bone resorption and regulating bone remodeling.

Macrophage colony-stimulating factor (M-CSF) binding to receptors (cFMS) present on osteoclast precursor cells stimulates their proliferation and inhibits their apoptosis [5]. Bone resorption is a primary event that occurs in response to mechanical stress via expression of receptor activator of NF-κB (RANK) ligand (RANKL). RANKL produced by differentiated osteoblasts binds to RANK on the surface of undifferentiated osteoclast precursor cells to induce differentiation resulting in osteoclastogenesis and eventually bone resorption [6]. These undifferentiated osteoclast precursor cells include bone marrow macrophages [7], splenocytes [8], and peripheral blood monocytes, and can be cultured in vitro to differentiate into osteoclasts in the presence of M-CSF and RANKL [9,10]. However, another factor, osteoprotegerin (OPG), which is produced by osteoblasts, stromal cells, and gingival and periodontal fibroblasts, inhibits bone resorption by binding to RANK, preventing RANK/RANKL interaction and osteoclastogenesis [6]. Thus, the RANKL/RANK/OPG axis is a primary pathway that mediates osteoclast differentiation.

The third type of bone cells are osteocytes, which are mature osteoblasts that are contained within the lacunae and secrete regulatory factors that influence osteoblast and osteoclast activity in response to mechanical stressors [1]. Osteocytes comprise 90–95% of total bone cells and are defined based on morphology and location rather than function during bone formation or resorption [11]. These cells are derived from mesenchymal stem cells through osteoblast differentiation and at the end of the bone formation cycle, a small subgroup of osteoblasts become osteocytes within the bone matrix [11].

The proliferation, differentiation, and activity of osteoblasts, osteoclasts, and osteocytes are mediated locally by autocrine and paracrine factors including growth factors, cytokines, and prostaglandins, as well as systemically by endocrine factors, such as parathyroid hormone (PTH), calcitonin, 1,25-dihydroxyvitamin D3, glucocorticoids, androgens, and estrogen [12,13,14,15,16]. Additional bone-derived autocrine factors, other than that of RANKL and OPG, are osteocalcin, sclerostin, and periostin. Osteocalcin is the most abundant osteoblast-specific, non-collagenous protein, and is a key determinant of bone formation. Sclerostin is a monomeric glycoprotein produced by osteocytes and articular chondrocytes, known to inhibit the Wnt/β-catenin signaling by binding to LRP5/6 receptors on osteoblasts [17,18,19]. This results in downstream inhibition of osteoblast differentiation, proliferation, and activity, leading to a decrease in bone formation [20]. Periostin is a highly conserved matricellular protein that regulates osteoblast function and bone formation via integrin receptors and Wnt/β-catenin signaling [21]. Periostin binds to integrins resulting in protein kinase B/Akt and FAK-mediated signaling activation, while GSK3β, a main regulator of the Wnt signaling pathway, is inhibited [22].

### 1.2. Irisin

The musculoskeletal system, comprised of skeletal muscle and bone, is highly involved in both the regulation of movement and metabolic health. Both tissues are involved in regulating whole-body metabolism aside from the basic mechanical role of bone and locomotion of muscle [23,24,25]. It is well established that increased skeletal muscle mass and strength improves bone mass and strength, and that a high correlation exists between the development of sarcopenia and that of osteoporosis [26,27,28,29].

Studies have clearly demonstrated that skeletal muscle can produce cytokines, termed myokines, which act in both an autocrine and endocrine manner. This was first demonstrated by Pederson and colleagues [30,31] with their studies on muscle-derived interleukin (IL)-6. Since this early work, many other myokines have been discovered, further suggesting that skeletal muscle communicates with other tissues via a family of myokines [32,33,34]. In addition to IL-6, the myokine myostatin, and muscle-derived growth factors such as insulin-like growth factor-1 (IGF-1), and fibroblast growth factors (FGF2 and FGF21) and irisin have been implicated in influencing bone homeostasis and metabolism [32].

Irisin was discovered in 2012 by Bostrom et al. [35] from their work on peroxisome proliferator-activated receptor gamma (PPARγ) coactivator-1 α (PGC1-α)-stimulated expression of fibronectin type III domain-containing protein 5 (FNDC5) that can be proteolytically cleaved to form irisin [35]. FNDC5 is a glycosylated type I membrane protein that contains an N-terminal signal peptide and fibronectin type III repeats. The C-terminal tail of FNDC5 remains within the cytoplasm, while the extracellular N-terminal portion is released as irisin [36,37]. In addition, irisin has been classified as a thermogenic protein that promotes energy expenditure and adipocyte browning via PGC-1α signaling [35] and uncoupling protein 1 (UCP1) expression [38,39].

Irisin was found to be expressed in skeletal muscle, adipose tissue, peripheral nerves, ovaries/testes, pancreas, liver, spleen, and stomach [40]. Administration of radioactively labelled irisin to mice revealed radiolabeling of the gallbladder, liver, and kidneys [41].

The irisin receptor belongs to a subset of integrin complexes, with the αV/β5 integrin complex receptor found to have the highest affinity for irisin [42]. Irisin-receptor binding results in downstream phosphorylation of tyrosine 397 on focal adhesion kinase (FAK), which in turn results in phosphorylation of threonine 308 on protein kinase B/Akt, and phosphorylation of cyclic AMP response element-binding protein (CREB) [42].

A number of reviews focusing on irisin have been published in the last 10–20 years [37,43,44,45,46,47,48,49,50]. The influence of exercise on FNDC5 mRNA expression and irisin levels is reviewed previously [43,51]. The effect of exercise on plasma irisin levels is not very clear. Some studies have reported significantly increased plasma irisin levels in mice with access to treadwheel running [35] and highly active individuals were shown to have increased circulating serum irisin levels, while individuals that live a less active and more sedentary lifestyle had reduced irisin levels [52]. Individuals undergoing endurance exercise doubled their irisin levels compared to control sedentary individuals [35]. However, several studies have shown that skeletal muscle FNDC5 mRNA expression was not increased with exercise, and instead exercise reduced irisin levels in men [53]. Furthermore, irisin and FNDC5 expression have been shown to be influenced by other factors including cold temperatures [54], and leptin [55].

The current review focuses on the role of irisin on bone homeostasis and all existing in vitro and in vivo animal and human studies are presented. A PubMed search was performed using the terms: irisin, bone, osteoblasts, osteoclasts, osteocytes, chondrocytes, bone-marrow progenitor cells, osteoporosis, osteopenia, ovariectomized, and osteoarthritis. These terms were searched using different combinations in order to ensure that all existing in vitro and in vivo animal and human studies were included. These studies are presented chronologically, and for easier access of main information, they are organized and presented in a table format.

## 2. Role of Irisin on Bone Homeostasis

### 2.1. Role of Irisin on Bone Homeostasis: In Vitro Evidence

#### 2.1.1. Osteoblasts

Colaianni et al. (2014) found increased FND5 mRNA and protein levels in primary cultures of muscle cells derived from mice exercised (wheel running) for 3 weeks (Table 1) [56]. The researchers used the media from the cultured muscle cells to treat bone marrow stromal cells isolated from mice and found induction of differentiation of the bone marrow stromal cells into osteoblasts as seen by the increased mRNA and protein levels of alkaline phosphatase (ALP) and collagen I alpha 1 (Col1a1) [56]. These data indicate that muscle cells derived from the exercised animals release myokines that have osteoblastogenic properties. Furthermore, the use of a specific irisin neutralizing antibody abolished the effects of the conditioned media, confirming that irisin is involved. The authors claimed that their data provide evidence that “Irisin directly targets osteoblasts, enhancing their differentiation” [56]. This claim would have been strongly supported if measurements of irisin levels in the media were performed, and with further experiments using recombinant irisin (r-irisin) examining its direct effects on osteoblasts.

Treatment of primary rat osteoblasts and pre-osteoblastic MC3T3-E1 cells with irisin (100 ng/mL) for 24 h resulted in increased proliferation and differentiation while MC3T3-E1 cells transfected with FNDC5/GFP expression lentivirus also showed increased proliferation [57]. Specifically, irisin treatment significantly increased the mRNA levels of osteoblast transcription regulators (Runx2, Osx) and early osteoblast differentiation marker genes (ALP, ColIa1) [57]. In addition, irisin treatment increased the mRNA levels of later differentiation genes (OC, OPG). Phosphorylated p38 and ERK protein levels in both primary rat osteoblasts and MC3T3-E1 cells were increased with irisin treatment and use of p38 (SB203580) or ERK (U0126) inhibitor reduced Runx2 mRNA level and ALP activity, in conjunction with inhibition of irisin-induced osteoblast proliferation and differentiation [57]. These data suggest that irisin may act via p38/ERK signaling to promote osteoblast proliferation and differentiation (Table 1).

Similarly, in another study, treatment of MC3T3-E1 cells with r-irisin (20 μmol/L) for 10 days resulted in increased ascorbic acid (AA)-induced osteoblast differentiation and increased mRNA levels of osteogenic markers including osterix (Osx), Runt-related transcription factor 2 (Runx2), and bone sialoprotein [58]. Chronic treatment (6 weeks) of MC3T3-E1 cells with irisin resulted in enhanced mineralization, suggesting improved osteoblast functionality [58]. Furthermore, nuclear β-catenin protein levels were increased in MC3T3-E1 cells treated with r-irisin for 6 h as compared to the control cells, while cytoplasmic β-catenin protein levels were reduced, suggesting that r-irisin may signal osteoblasts directly, and increase osteoblastogenesis via β-catenin signaling activation (Table 1) [58].

Treatment of mouse bone marrow cells with irisin dose-dependently suppressed M-CSF and RANKL-stimulated osteoclast formation and significantly reduced primary osteoblast RANKL mRNA levels and the ratio of RANKL to OPG mRNA [59]. Furthermore, FNDC5 mRNA levels were increased in C2C12 cells treated with fluid flow shear stress (FFSS), a biological important inducer of mechanical stress [59].

Treatment of osteoblastic MC3T3E1 cells and bone marrow stromal cells with r-irisin (10 nM) for 48 h resulted in increased proliferation and osteoblastic differentiation [60]. R-irisin treatment increased Runx2, and Osx and osteoblastic differentiation markers ALP and Col1a1 mRNA levels. Additionally, r-irisin treatment increased the formation of ALP-positive colonies in bone marrow stromal cells [60]. These effects are preferentially through aerobic glycolysis, as r-irisin increased lactate dehydrogenase A (LDHA) and pyruvate dehydrogenase kinase 1 (PDK1) protein levels and increased lactate serum levels. In the presence of dichloroacetate, r-irisin-mediated aerobic glycolysis was prevented [60]. These data suggest that r-irisin regulates osteoblast differentiation and proliferation via aerobic glycolysis (Table 1).

In a study by Zeng et al., point mutation of key regions, Glu-57 and Ile-107, and His-tag addition on the C-terminal of irisin, resulted in reduced activity of r-irisin (20 nM; 14 days) on osteogenesis and proliferation of MC3T3E1 cells [61]. The effect of r-irisin on ALP, Col1a1, and Runx2 mRNA expression was reduced by 90.1%, 88.8%, and 96.6% with irisin E57K, irisin I107F, and irisin-C-His, respectively. Additionally, salt bridge, Arg75, point mutation, only partially reduced the activity of r-irisin on ALP, Col1a1, and Runx2 mRNA expression, while N-terminal His-tag addition had no effect [61]. These studies indicate that the residues 55–58, and 106–108 and the C-terminal region are necessary for irisin activity to regulate key osteogenesis mRNA levels, but it is not known whether they influence irisin binding to its receptor (Table 1).

Treatment of MC3T3-E1 osteoblasts with r-irisin (100 ng/mL) for 8 h reduced parathyroid hormone receptor mRNA level. Treatment of C2C12 myotubes with the parathyroid hormone analogue teriparatide (10 nM) for 8 h resulted in significantly reduced FNDC5 expression and increased phosphorylated ERK1/2 protein levels (Table 1) [62]. Postmenopausal women with primary hyperparathyroidism had significantly lower serum irisin levels than postmenopausal women without hyperparathyroidism [62]. Although this study indicates an interplay between irisin and parathyroid hormone, more studies are required to examine this relationship in depth.

Treatment of bone marrow mesenchymal stem cells with irisin (40 µM) for 48 h resulted in increased osteogenic differentiation and autophagy [63]. Irisin treatment increased microtubule-associated protein light chain 3 (LC3)-I/II and autophagy related 5 (Atg5) mRNA levels, indicating increased autophagy. The use of bafilomycin A1 and Atg5 small interfering RNA strategy prevented the increase in LC3 and Atg5 mRNA levels by irisin [63]. In addition, irisin increased osteogenic differentiation, osteogenesis genes (Runx2, ALP, and OCN) and Wnt/β-catenin signaling (β-catenin, Lef-1, and Tcf-4) mRNA levels, and calcified nodules [63]. Therefore, these data suggest that irisin may stimulate bone marrow mesenchymal stem cell autophagy and osteogenic differentiation via the Wnt/β-catenin signaling cascade (Table 1).

Osteoblasts, osteoclasts, and endothelial cells were seeded on Skelite discs and exposed to microgravity for 14 days in the eOSTEO hardware to mimic the bone microenvironment in microgravity aboard the SpaceX Dragon cargo ferry to the International Space Station [64]. Treatment with r-irisin (100 ng/mL) during the 14 days of orbit resulted in increased osteoblast differentiation and reduced osteoclastogenesis. Irisin treatment increased the mRNA levels of genes encoding osteoblast transcription factors (Atf4, Runx2, Osterix) and osteoblast proteins (Collagen I, osteoprotegerin) [64]. These data suggest that irisin treatment promotes osteoblast differentiation in microgravity and may prevent bone loss in astronauts during space expeditions (Table 1).

Macrophages play a role in bone healing and Ye et al. [65] examined the effect of irisin on macrophage polarization and osteogenesis. Treatment of Raw264.7 macrophages with irisin (200 ng/mL) for 48 h resulted in increased polarization of M0 and M1 macrophages towards M2 phenotype (Table 1) [65]. In addition, irisin treatment increased the expression of CD206-APC ARG-1 and TGF-β1 and reduced the expression of CD86-PE and TNF-α. In addition, co-culture of pre-osteoblast MC3T3-E1 cells with irisin-treated Raw264.7 macrophages resulted in increased osteogenesis, mineralized particle formation, and AMPK activation [65]. Knockdown of AMPK-α by siRNA completely abrogated the irisin-induced effect on macrophage polarization and the osteogenic ability of Raw264.7 macrophages [65]. These data suggest that irisin acts via AMPK activation to induce macrophage polarization and enhance osteogenesis in osteoblasts.

The study of the effects of irisin on osteoblasts is still in its relative infancy. The existing studies have suggested that irisin may help to promote osteoblastogenesis. Mechanistically, irisin increases osteoblast proliferation, differentiation, and mineralization via the activation of the Wnt and p38/ERK MAPK signaling pathways, both resulting in increased expression of RUNX2, an important osteogenic marker, and downstream activation of ALP and OC (Figure 1). Future experiments should focus on further elucidating the mechanisms of irisin-induced osteoblastogenesis and promotion of bone growth. Additionally, many studies have only examined the effects of irisin on osteoblasts under healthy conditions, and future studies should focus on the effect of irisin on osteoblast under conditions of bone loss or bone stress to further determine the role of irisin on bone homeostasis.

#### 2.1.2. Osteoclasts

Zhang et al. (2017) examined the effect of r-irisin on RANKL-induced osteoclast differentiation [58]. Treatment of pre-osteoclastic RAW264.7 cells with r-irisin (20 nmol/L) for 3 days resulted in significantly reduced RANKL-induced osteoclast differentiation markers tartrate-resistant acid phosphatase (TRAP) and cathepsin K (CK) mRNA levels. In addition, r-irisin treatment decreased RANKL-induced TRAP-positive multinucleated cells and nuclear factor of activated T cell (NFATc1) mRNA and protein levels [58]. Treatment of RAW264.7 cells with r-irisin for 60 min reduced protein levels of RANKL-induced calcineurin and phosphorylated JNK and AKT (Table 2) [58]. These data suggest that r-irisin treatment significantly reduces osteoclastogenesis via reduced NFAT1c signaling and TRAP/CK differentiation marker expression and suggest that irisin could be utilized as a potential therapeutic option to prevent osteoclast generation in order to promote bone remodeling.

Treatment of primary hematopoietic progenitor cells and pre-osteoclastic RAW264.7 macrophage cells with irisin (2–20 ng/mL) for 4 h, 24 h, or 7 days resulted in significantly increased osteoclast number and size dose- and time-dependently [67]. In the presence of a neutralizing antibody, irisin treatment was unable to increase osteoclast number, indicating that the αV/β5 integrin receptor complex is necessary for the effects of irisin on osteoclasts. In addition, irisin treatment increased the mRNA level of the irisin αV/β5 integrin receptor. In additional experiments irisin treatment increased resorption pit area and osteoclast number, suggesting effects on bone resorption [67]. Resorption as a percent of the total was increased with irisin treatment; however, there was no change when resorption was normalized as a percent of osteoclasts. However, culturing the cells using resorbable calcium phosphate substrate or collagen substrate resulted in significantly increased resorption, as a percent of the total area, by irisin treatment [67]. In addition, irisin treatment increased mRNA levels of resorption markers, (Adamts5 and Loxl3), secreted clastokine markers (Postn, Igfbp5, Tgfb2, and Sparc), key differentiation markers (Atp6vod2, Cfos, Dcstamp, Fam102a, Itgb3, Nfatc, Nrf2, Rank, Rela, and Rgs12), and macrophage markers (Mst1r and Itgax) and reduced mRNA levels of lymphocyte markers (Slamf8 and H2-aa) (Table 2) [67]. The difference seen in the study by Estell et al., which provides evidence of irisin promoting osteoclastogenesis [67], and the study by Zhang et al., which demonstrates inhibition of osteoclastogenesis [58], may be due to temporal differences, with short duration irisin treatment (7 days) promoting osteoclastogenesis [67], and chronic irisin treatment (2 months) [58], reducing osteoclastogenesis via increased activity of the Mck promoter of Fndc5.

Treatment of Raw264.7 cells and mouse bone marrow monocytes with irisin (20 nM) for 4 days resulted in increased osteoclast precursor cell proliferation, but reduced osteoclast differentiation [68]. In addition, irisin treatment reduced osteoclast differentiation marker genes, receptor activators of NF-kB, NFATc1, cytoplasmic 1, CK, and TRAP mRNA levels and reduced the number of TRAP-positive multinucleated cells [68]. Treatment of pre-osteoclast cells with irisin for 60 min also significantly reduced IκBα and phosphorylated p65 protein levels. Irisin treatment had differential effects on phosphorylated JNK and p38 protein levels, as RANKL-induced JNK phosphorylation was delayed and did not reach the same peak level seen with RANKL alone [68]. Phosphorylation of p38 was increased with irisin treatment and remained phosphorylated/activated for longer, suggesting that p38 and JNK may be specifically activated by irisin. Pre-osteoclast Raw264.7 cells pretreated with p38 inhibitor (SB203580; 20 μM) or JNK inhibitor (SP600125; 40 μM) followed by treatment with irisin resulted in reduced mRNA levels of osteoclast differentiation marker (Table 2) [68]. These data suggest that irisin reduces osteoclast differentiation by modulating important differentiation marker gene expression independent of JNK/p38 signaling.

Treatment of primary BMSCs isolated from Osx-Cre:FNDC5/irisin knockout (F/I KO) mice with irisin (4 and 20 μmol/L) for 5 days increased osteoblastogenesis and mineralization and reduced osteoclastogenesis [69]. Treatment with irisin increased amino acid (AA)-stimulated osteogenic markers Runx2, Satb2, Bsp, Col1a1, and ALP mRNA levels. Cytosolic β-catenin protein levels were reduced with 6 h of irisin treatment, while nuclear β-catenin protein levels increased [69]. Treatment of RANKL-induced primary pre-osteoclast macrophages with irisin resulted in significantly reduced osteoclastogenesis markers (Trap, MMP9, and NFATC1) mRNA levels and NFATC1 protein levels. In addition, phosphorylated Akt1, p38, and calcineurin protein levels, TRAP staining, and TRAP-positive multinucleated cell numbers were reduced with irisin treatment [69].

The above studies indicate that treatment of pre-osteoclast cells with irisin reduced their differentiation, and production of key osteoclast proteins TRAP and CK. Mechanistically, irisin reduced RANKL-induced osteoclast differentiation and NFAT1c transcriptional regulation of specific differentiation genes (Figure 2). However, the direct effects of irisin still remain to be elucidated and requires additional studies to fully understand how irisin reduces osteoclast differentiation. In addition, as was stated with osteoblast cells, the studies have only examined the effects of irisin on normal osteoclast differentiation, and future studies to determine the potential of irisin as a regulator of bone homeostasis should focus on the effect of irisin on osteoclast differentiation under conditions of bone loss or bone stress.

#### 2.1.3. Osteocytes

In a study by Kim et al. (2018), treatment of osteocyte-like (MLO-Y4) cells with physiologically relevant concentrations (1–500 ng/mL) of irisin for 16 h resulted in significantly reduced hydrogen peroxide (H_2_O_2_)-induced apoptosis, suggesting that irisin can block osteocyte cell death [42]. Incubation of MLO-Y4 cells with antagonistic antibodies against integrin αV/β5 completely blocked the irisin-mediated phosphorylation of FAK, CREB, and Zyxin. Irisin binds to integrin αV/β5 via a loop that is structurally similar to RGD motifs [42]. Use of RGD peptide suppressed the irisin-induced phosphorylation of FAK, CREB, and Zyxin. Additionally, the irisin-induced effects were blocked with the addition of integrin αV inhibitors echistatin, cyclo RGDyK and SB273005 (Table 3) [42].

Exposure of MLO-Y4 osteocyte cells to irisin (100 ng/mL) for 24 h resulted in significantly reduced cyclic stretching-induced apoptosis and increased osteocyte proliferation [70]. Irisin treatment significantly reduced sclerostin mRNA level and increased the ratio of OPG/RANKL, suggesting improved bone remodeling. In addition, irisin increased phosphorylated Erk and p38 protein levels. However, in the presence of small molecule inhibitors, only ERK inhibition (PD98059; 10 μM) prevented the irisin-mediated decrease in osteocyte apoptosis, suggesting that the anti-apoptotic effects of irisin are mediated by ERK activation (Table 3) [70].

In a study by Storlino et al. (2020), treatment of MLO-Y4 cells with r-irisin (100 ng/mL) for 24 h completely prevented the H_2_O_2_-induced apoptosis (Table 3) [71]. Irisin treatment significantly reduced H_2_O_2_-induced caspase-9 protein level and dexamethasone-induced cleavage of caspase-3. Treating of MLO-Y4 cells with r-irisin for 8 h also significantly increased Tfam mRNA levels and the pro-survival Bcl2/Bax ratio [71]. Additionally, irisin treatment increased phosphorylated ERK protein levels and rapidly increased transcription factor Atf4, a known transcriptional regulator of bone development, while sclerostin and Dickkopf WNT signaling pathway inhibitor 1 (DKK1) mRNA levels were reduced. Pretreatment of MLO-Y4 cells with an ERK inhibitor (PD98059; 50 uM) prevented the r-irisin-induced increased of Atf4 mRNA level [71]. These data suggest that r-irisin treatment helps to prevent stress-induced apoptosis via reduced caspase-9 and caspase-3 signaling and increased Atf4 transcriptional activity. However, the direct effect of Atf4 on apoptosis regulation remains to be elucidated; further experiments utilizing inhibitors or knockout models are necessary to determine if irisin acts via Atf4 to restore osteocyte numbers and bone development.

The above studies indicate that treatment of osteocytes with irisin prevented the H_2_O_2_- and cyclic stretching-induced apoptosis, and increased ATF4 expression via β-catenin/AP1 and AKT/ERK signaling (Figure 3). However, the direct effects of irisin still remains to be elucidated and requires additional studies to fully understand how irisin protects against apoptosis and increases osteocyte proteins known to be involved in bone development. In addition, future studies should focus on understanding the potential of irisin as a regulator of bone homeostasis under conditions of bone loss/stress.

#### 2.1.4. Chondrocytes

Treatment of human osteoarthritic chondrocytes (hOAC) with human r-irisin (25 ng/mL) for 7 days resulted in increased cell proliferation and production of glycosaminoglycan (GAG), an important extracellular matrix component of cartilage tissue (Table 4) [72]. Treatment with irisin significantly decreased inflammatory marker (IL-1, IL-6, iNOS, MMP-1, and MMP-13) mRNA levels while increased mRNA levels of anti-catabolic enzymes, tissue inhibitor of matrix metalloproteinase (TIMP)-1 and TIMP-3. Type II collagen mRNA and protein levels were increased with irisin treatment. In addition, phosphorylated p38, Akt, JNK, and total NF_K_B protein levels were significantly reduced with irisin treatment [72]. These data suggest that irisin may promote chondrocyte growth and GAG production via inhibition of p38, Akt, JNK, and NF_K_B signaling and suggest the potential use of irisin towards the treatment of osteoarthritis.

Primary mouse chondrocytes isolated from mice with destabilized medial meniscus (DMM)-induced osteoarthritis had significantly reduced FNDC5 and LC3 expression and weak immunoreactivity of the proliferating cell nuclear antigen (PCNA) [73]. Treatment of chondrocytes with irisin (10 ng/mL) for 6 h attenuated these DMM-induced effects and significantly increased FNDC5 and LC3 expression and PCNA immunoreactivity. Additionally, chondrocytes incubated with IL-1β (5 ng/mL) for 72 h had reduced Alcian blue-stained GAG and ECM production [73]. Irisin treatment attenuated these IL-1β-induced effects and dose-dependently increased ECM accumulation. In addition, irisin treatment for 24 h increased FNDC5 expression, cell growth, and chondrocytic markers (collagen II, aggrecan, and Sox9) mRNA levels and significantly reduced synovitis-promoting factors (MMP9 and CEGF) mRNA levels in the presence of IL-1β [73]. Treatment of IL-1β-induced inflamed chondrocytes with irisin significantly increased PGC-1α and Tfam mRNA levels, ATP production, mitochondrial membrane potential, mitophagy, and autophagosome formation. Irisin treatment also attenuated the IL-1β-mediated reactive oxygen radical production and increased Sirt3 and UCP-1 protein levels [73]. These data suggest that irisin has a protective role against chondrocyte dysfunction by repressing autophagy and mitochondria dysfunction and improving survival and anabolism of DMM- and IL-1β-induced inflamed chondrocytes (Table 4).

### 2.2. Role of Irisin on Bone Homeostasis: In Vivo Evidence

In a study by Colaianni et al. (2015), injection of young male mice with r-irisin (100 µg/kg^−1^) once a week for 4 weeks resulted in increased both cortical bone mass and strength (Table 5) [74]. Using X-ray imaging, irisin injection increased the radiodensity of the femoral and tibia bones and bone perimeters including total cross-sectional area and marrow area. Interestingly the trabecular bone radiodensity was unaffected with irisin injection. Irisin treated mice had significantly increased polar moment of inertia, an objects ability to resist torsion, and bending strength [74]. These data suggest that irisin may be involved in regulating both bone mass and strength (Figure 4).

Injection of wild-type mice with r-irisin (1 mg/kg b.w.) for 6 days resulted in increased mRNA levels of sclerostin, a bone remodeling regulator, in osteocyte-enriched bones and increased sclerostin protein in plasma [42]. Co-injection of mice with Cyclo-RGDyK (cRGDyK; 1 mg/kg), an αV inhibitor/antagonist, or SB273005, an integrin inhibitor for αV/β5 and αV/β, prevented the irisin-induced effects on sclerostin, suggesting that the integrin αV/β5 is important for irisin function in osteocytes [42].

Using a transgenic mouse model with skeletal muscle-specific forced expression of FNDC5, Estell et al. (2020) found the mice to have lower cortical bone area at younger ages than the wild type. The transgenic line had significantly lower trabecular bone volume fraction and cortical thickness. In mesenchymal cells harvested from transgenic and wild type mice there was no difference in genotype, but the transgenic mice had significantly higher number and size of osteoclasts (Table 5) [67].

Luo et al. [75] generated global irisin knockout mice, and found reduced bone strength and bone mass compared to control animals [75]. Additionally, osteoclast number and RANKL cell surface expression were increased in irisin-lacking mice. The mice also appeared to have a poor browning response, with increased intraperitoneal white adipose cell size and reduced number of brown adipose cells within the interscapular tissue. This was accompanied with disordered metabolism in mice lacking irisin as they presented with hyperlipidemia and insulin resistance, reduced HDL-cholesterol level, and increased LDL-cholesterol level (Table 5) [75]. These data suggest that irisin may be involved in the regulation of glucose/lipid metabolism and is necessary for bone homeostasis (Figure 4).

In a study by Zhu et al., FNDC5/irisin deletion within the osteoblast lineage mice (Osx-Cre:FNDC5/irisin KO mice) demonstrated reduced irisin mRNA and protein levels in bone, reduced bone density, and delayed bone development and mineralization [69]. The reduction in mineralization and bone development was shown at 6, 10, and 18 weeks. These irisin-knockdown mice also presented with lower cortical bone mineral density and trabecular bone/tissue volume as compared to the controls, but the cortical bone surface/volume ratio increased suggesting thinner cortical bones [69]. Osx-Cre:FNDC5/irisin KO mice had reduced osteoblast-related genes (Runx2, Bsp, Osx, and Alp) mRNA levels and increased osteoclast-related genes (cathepsin K, Mmp9, and Trap) mRNA levels. In addition, the protective effects of exercise, including increased bone strength and body weight loss were reduced with irisin deficiency and were enhanced with administration of r-irisin during exercise for 14 days [69]. Cultured MSCs from Osx-Cre:FNDC5/irisin KO mice had reduced osteoblastogenesis and increased osteoclastogenesis. Irisin treatment reduced these effects by stimulating osteogenesis and inhibiting osteoclastogenesis (Figure 4) [69].

In mice that underwent 2 weeks of voluntary wheel-running exercise training, bone and articular cartilage tissue FNDC5 and irisin mRNA and protein levels were significantly increased [58]. In addition, FNDC5 and irisin protein levels were increased in the growth plate, trabecular bone, cortical bone, articular cartilage, and muscle-bone interface with exercise (Table 5) [58]. These data suggest that irisin is produced in bone in response to exercise.

In a study by Kawao et al., administration of irisin to mice with hindlimb unloading and sciatic neurectomy resulted in reduced muscle volume at the tibia, soleus, and gastrocnemius muscle weight and trabecular bone mineral density of the tibia (Table 5) [59]. Additionally, hindlimb unloading and sciatic neurectomy reduced soleus muscle FNDC5 mRNA levels. Regression analysis indicated that soleus muscle FNDC5 mRNA levels were positively correlated to tibia trabecular bone marrow density and FNDC5 mRNA levels were negatively associated with tibia RANKL mRNA levels [59]. These data suggest that mechanical unloading reduces skeletal muscle irisin levels and highlights the importance of irisin in regulating muscle and bone homeostasis in the presence of mechanical stress in mice (Figure 4).

Hindlimb unloaded mice, as a model to stimulate microgravity effects on bone in vivo, had increased femur bone loss, and reduced mineral apposition rates of trabecular and cortical bone (Table 5) [76]. Additionally, Fndc5 mRNA levels and mRNA levels of the osteogenesis markers (Alp, ColIa1) were reduced. In addition, microgravity was simulated in primary osteoblasts via rotation along two independent axes to change the orientation in random modes [76]. The ALP activity and mRNA levels of Alp, ColIa1, and Fndc5 of primary osteoblasts were reduced with the simulated microgravity. Treatment of primary osteoblasts that underwent simulated microgravity with r-irisin (1 nM) for 14 days significantly increased the mRNA levels of osteogenic markers (Alp and ColIa1), increased ALP activity, and mineralization [76]. In addition, irisin treatment increased mRNA levels of (CyclinA2, CyclinD1, CyclinE1, CDK2, and CDK12) and significantly promoted osteoblast proliferation and differentiation under simulated microgravity conditions. Additionally, r-irisin treatment increased β-catenin mRNA levels [76]. These data suggest that r-irisin positively stimulates osteoblast differentiation and may help to prevent bone loss induced by microgravity.

Administration of r-irisin (18 ng/mL, three times per week) to hindlimb unloaded male Sprague–Dawley rats for 8 weeks resulted in improved bone homeostasis and increased bone formation rate [77]. Exogenous r-irisin treatment significantly reduced osteoclast surfaces and the secretion of TNF-α, IL-17, RANKL, and sclerostin by osteocytes in the unloaded hindlimb, while only minimal changes occurred in the humerus (Table 5) [77]. These data showed that irisin administration significantly reduced the pro-inflammatory state associated with unloaded hindlimb and may protect from disuse-induced bone loss (Figure 4).

Storlino et al. treated hindlimb unloaded C57BL6 mice with r-irisin (100 µg/kg) once a week for 4 weeks to examine the effects on osteocyte apoptosis (Table 5) [71]. Irisin injections rescued the caspase upregulation that was seen in the bone tissue disuse model. When the rear legs of the mice were suspended to prevent their use, there was a significant increase in the expression of pro-apoptotic genes caspase-9 and caspase-3 and the Bcl2/Bax ratio was increased. All of these effects were reversed with irisin administration [71].

Global FNDC5-null mice had significantly reduced mRNA levels of apoptosis regulator gene RANKL; however, RANKL receptor, OPG, was not altered. The FNDC5-null mice also had increased femoral trabecular bone mass and more connectivity density. Interestingly, FNDC5-null mice were resistant to ovariectomy (OVX) and had no bone loss or change in RANKL mRNA level, and a reduction in bone resorption. When compared to the control wild type group, FNDC5-null mice lacked OVX-induced osteocytic osteolysis. In control wild type mice, plasma irisin levels, 2 weeks post-OVX, were increased 2.4-fold (Table 5) [42]. These data suggest that irisin specifically acts through the αV/β5 integrin receptor in osteocytes. In the absence of estrogen due to OVX, FNDC5-KO mice had increased bone protection from osteoporosis by preventing osteoclast proliferation and eroded surfaces (Figure 4). The authors suggest that the mechanism of action of irisin may be similar to that of the parathyroid hormone, which acts both to stimulate resorption and acts anabolically when administered intermittently. Clearly, more studies are necessary to elucidate the effects of irisin in OVX and osteoporosis.

Intraperitoneal administration of OVX mice with r-irisin (100 µg/kg, twice a week) for 5 weeks prevented trabecular bone loss and increased bone microarchitecture [78]. Irisin administration significantly increased bone marrow density, bone volume to tissue volume ratio, connection density, bone stiffness, and trabecular number parameters. In addition, r-irisin-treated OVX mice had increased number of osteoblasts on the trabecular surface and reduced number of osteoclasts [78]. Osteocalcin serum level was increased and TRAP concentration in serum was reduced with r-irisin administration (Table 5) [78]. These data suggest that irisin helps to preserve bone microarchitecture and improve osteoblast number and activity in the absence of estrogen (Figure 4).

Single-dose administration of irisin (1 mmol/L) to postmenopausal Sprague–Dawley rats with osteoporosis resulted in increased trabecular thickness, number, and bone mineral density and reduced osteoblast apoptosis (Table 5) [79]. Serum ALP level, caspase-2 and NLRP3 mRNA and protein levels were reduced with irisin treatment, while Runx2, OC, Bcl-2, and Nrf2 mRNA levels and Bcl-2 and Nrf2 protein levels were increased [79]. These data suggest that treatment of postmenopausal osteoporotic rats with irisin reduces the NLRP3 inflammasome to prevent osteoblast apoptosis and protect against bone loss (Figure 4).

OVX mice that underwent moderate intensity treadmill exercise had increased irisin protein and Fndc5 mRNA levels in gastrocnemius and soleus muscle [80]. Gastrocnemius Fndc5 mRNA levels were positively associated with trabecular bone marrow density, but not cortical bone marrow density at the femurs and tibias. In addition, treadmill exercise increased gastrocnemius muscle PGC1α mRNA levels [80]. These data suggest that chronic exercise promotes increased skeletal muscle irisin expression in estrogen-deficient mice, and that this may help to protect or increase trabecular bone marrow density (Figure 4).

Intraperitoneal injection of r-irisin (100 ug/kg b.w.) to androgen deficient and osteopenic mice once a week for 8 weeks resulted in significantly improved trabecular bone marrow density; however, muscle mass was not improved (Table 5) [81]. Additionally, irisin gene Fndc5 mRNA levels was significantly reduced by androgen deficiency in the soleus and gastrocnemius muscle. Soleus muscle Fndc5 mRNA levels positively correlated with trabecular bone marrow density, but not in the cortical [81]. Therefore, Fndc5 gene expression in the muscle may positively influence bone structure impaired by androgen deficiency (Figure 4).

Inflammatory bowel disease is a chronic disease that is associated with gastrointestinal dysfunction and inflammation-induced bone loss [82]. Intraperitoneal injection of irisin (18 ng/mL; 2 times per week) for 3 weeks to Sprague–Dawley rats, that had inflammatory bowel disease induced by 2,4,6-trinitrobenzenesulfonic acid; 30 mg/kg b.w. resulted in improved gut and bone outcomes via reduced inflammation and restored structure [82]. Irisin administration reduced lymphatic hyperproliferation and osteoclast-covered bone surface area and increased bone formation rate. In addition, irisin administration significantly reduced gut and osteocyte TNF-α and RANKL protein levels to the level of control (Table 5) [82]. These data suggest that irisin may mitigate both local inflammation and distant changes in bone in rats with inflammatory bowel disease (Figure 4).

Sprague–Dawley rats with severe inflammatory bowel disease were administered r-irisin (18 ng/mL b.w.; twice a week) for 3 weeks had reduced dextran sodium sulfate (DSS)-stimulated colon inflammation and increased bone formation (Table 5) [83]. Irisin administration reduced osteoclast surface expression and pro-inflammatory (TNF-α, RANKL, OPG, IL-6, and sclerostin)-positive osteocytes. However, irisin was unable to improve bone density or bone mechanical properties induced by DSS [83]. These data suggest that irisin could potentially be utilized as an anti-inflammatory treatment and can improve gut and bone homeostasis in the presence of severe inflammatory bowel disease (Figure 4).

A mouse model of knee osteoarthritis was created surgically by injecting chloral hydrate into the anterior cruciate ligament transection (ACLT) which caused a decrease in the tibial hyaline cartilage and an increase in calcification. This was reversed however in the group of ACLT mice that were injected with r-irisin at a daily dose of 100 µg/kg b.w. for 4 weeks (Table 5) [70]. Microtomography suggested that the irisin injections rescued the damaging effects of ACLT in the subchondral and in the tibial trabecular bone as confirmed with bone histomorphometry staining. The subchondral bone area had less ACLT-stimulated TRAP+ stained osteoclasts in the irisin group as compared the vehicle control group (Figure 4). Immunohistochemical staining showed decreased expression of MMP-13 and caspase 3 suggesting that irisin was able to decrease osteocyte apoptosis and cartilage degradation after ACLT [70].

Wang et al., using an established model of destabilization of medial meniscus (DMM) in mice, showed that intra-articular irisin injections were able to rescue the damaged knee joints [73] and less cartilage destruction and synovitis was observed. Irisin injection also rescued the walking pattern of the DMM-injured mice. Within the four measures of walking pattern examined, there was a significant difference between the control and the DMM group of mice, while there was no significant difference between the control and the DMM-irisin group suggesting walking pattern can be rescued with irisin treatments. Chondrocytes derived from DMM-injured mice showed less FNDC5 and LC3 with immunostaining but higher levels of proliferating cell nuclear antigen and TUNEL and all of these findings were reversed by irisin treatment (Table 5) [73].

A study using rats showed that animals fed a high fat diet (HFD) had a significant increase in weight, over a group fed a control diet, while the amount of food intake remained the same (Table 5) [84]. The high fat diet also led to a decrease in BMD and changes in the microstructure of the femurs and tibia. With the introduction of swimming exercise at 8-weeks the results to the BMD and microstructure changes were reversed suggesting exercise is effective at improving bone health. Introducing exercise to the high fat diet group lead to an increase in the β-catenin, PGC-1α, and FNDC5 levels that were decreased with the HFD alone. HFD decreased the serum levels of OC and irisin but these decreased were rescued with the introduction of exercise. These results suggest that exercise can improve bone accrual attenuation, and that irisin may be involved in the mechanism [84].

### 2.3. Role of Irisin on Bone Homeostasis: Clinical Evidence

Serum levels of irisin in humans have been measured [42,46,85,86]. Irisin levels in a sedentary individual are approximately 3.6 ng/mL, and are significantly increased to 4.3 ng/mL with exercise (Table 6) [85]. Additionally, Moreno et al. reported even higher serum levels of irisin in physically active individuals, reaching levels of μg/mL [86]. Importantly, the half-life of irisin in mice injected with r-irisin for 6 days was found to be ~30 min [42].

In a study by Colaianni et al., healthy children demonstrated a positive association between irisin levels and bone mineral content [87]. In addition, irisin levels positively correlated with osteocalcin and bone resorption marker CTX, and negatively correlated with DKK-1 serum levels (Table 6) [87]. These data suggest that irisin may be a determinant of bone mineral status and could be necessary for bone formation during childhood (Figure 5).

Adolescent male swimmers showed a post-swimming increase in the anti-inflammatory interleukin IL-10 while adults had an increase in IL-6, IL-β, IL-10, and TNF-α. Adults showed a decrease in irisin level after swimming exercise, but the adolescent group did not show a change in irisin level. These results suggest adolescent athletes may have a blunted inflammatory and myokine response following exercise which may explain why young athletes require less recovery time from intense exercise (Table 6) [88].

In another study by Colaianni et al., athletic Caucasian football players of the Bari team (Bari, Italy), had a positive correlation between irisin and total body bone mineral density and bone mineral content (Table 6) [89]. In addition, linear association was observed between irisin and bone mineral density of different bone locations, including right arm, lumbar vertebrae, and head (Figure 5) [89].

In women that are postmenopausal and with low bone mass, serum irisin levels were inversely correlated with age, parathyroid hormone, and creatinine levels, and were reduced in women with previous osteoporotic fractures [90]. Additionally, in multiple linear regression, osteoporotic fracture occurrence and parathyroid hormone levels were inversely associated with irisin levels (Table 6) [90]. Therefore, previous osteoporotic fractures can be utilized as a negative predictor of circulating irisin levels in postmenopausal women with low bone mass (Figure 5).

In a study by Engin-Ustun et al., serum irisin levels were significantly reduced in women with postmenopausal osteoporosis [91]. In addition, chemerin serum levels were reduced in women with osteoporosis, while c-reactive protein levels were increased, suggesting that adipokines, irisin and chemerin, may help to reduce the pathogenesis of osteoporosis (Table 6) [91].

An extreme sampling design was implemented by Wu et al., who utilized a large screened Chinese elderly population with extremely high hip bone marrow density (Table 6) [92]. Plasma irisin levels were significantly elevated in these subjects when compared to low bone marrow density controls in both discovery and validation samples. However, this was sex specific with the increased levels of irisin present only in males and not females. Additional correlation analyses demonstrated increased plasma irisin levels correlated with bone marrow density and triglyceride levels in Chinese elderly males (Figure 5) [92].

In a case-control study by Zhang et al., geriatric Chinese men with osteoporosis or osteopenia had reduced serum irisin levels compared to age-matched controls without osteoporosis or osteopenia [93]. In addition, multiple linear regression analysis showed that serum irisin levels are an independent factor for bone marrow density (Table 6) [93]. These data suggest that irisin may impact bone marrow density in geriatric Chinese men (Figure 5).

In patients with rheumatoid arthritis (RA), serum irisin level was significantly reduced and was associated with increased low-fracture bone fractures in the anamnesis [94]. Additionally, patients with RA had higher RA activity degree (DAS28), extra-articular manifestations, longer duration of disease, increased function joint failure and reduced serum levels of vitamin D (Table 6) [94]. These data suggest that serum irisin levels may be a predictive biomarker for bone fractures in patients with RA (Figure 5).

Osteoarthritic bone tissue cells were isolated from patients with end-stage osteoarthritis and demonstrated increased cartilage damage. In addition, chondrocytes isolated from the knee of patients with osteoarthritis had increased 8-OHdG production and TUNEL expression, indicating increased oxidative stress and DNA fragmentation (Table 6). While, autophagosome marker LC3-II and FNDC5 surface expression were downregulated in articular cartilage [73]. These data suggest that patients with osteoarthritis are more prone to bone cell apoptosis and that irisin/FNDC5 may be a biomarker for chondrocyte protection.

Serum irisin levels were significantly reduced in elderly Chinese women that had suffered minimal trauma hip fractures compared to women without fractures [95]. In addition, serum insulin levels was an independent variable for bone marrow density, however low serum concentrations of irisin was associated with increased risk of hip fracture, suggesting that normal to high levels of circulating irisin is necessary for improved bone marrow density and the protective effects on bone that is associated with irisin (Table 6; Figure 5) [95].

Individuals undergoing total hip or knee replacement and with osteopenia/osteoporosis had reduced serum irisin levels compared to healthy controls (Table 6) [96]. In addition, serum irisin levels negatively correlated with age, and positively correlated with femoral and vertebral bone marrow density. In muscle biopsies, irisin positively correlated with Fndc5 mRNA levels, and in bone biopsies, irisin positively correlated with osteocalcin mRNA levels [96]. In skeletal muscle, FNDC5 positive fibers positively correlated with total femur and femoral neck bone marrow density. Researchers further confirmed the influence of age with serum irisin levels in vitro. Treatment of pre-osteoblast-like MC3T3-E1 cells with r-irisin (100 ng/mL) for 24 h resulted in significantly reduced senescence marker p21 mRNA level [96]. These data suggest the irisin levels are associated with increased protection against age-related osteoporosis and osteoblast aging progression.

In a cross-sectional study by Lu et al., individuals with osteoporosis and osteopenia and on maintenance hemodialysis demonstrated a positive association between lumbar bone marrow density and serum irisin levels [97]. Additionally, lumbar T-score was negatively associated with serum irisin levels, indicating that irisin may have a positive effect on bone density (Table 6; Figure 5) [97].

In a cross-sectional study by Palermo et al., overweight individuals with a previous vertebral osteoporotic fracture had reduced serum irisin levels (Table 6) [98]. The lower irisin levels remained significant after adjustments for creatinine, vitamin D, lean muscle mass, lumbar bone marrow density, and femoral bone marrow density. However, irisin serum levels were not correlated with bone marrow density, or daily physical activity [98]. These data suggest that irisin levels may be influenced by vertebral fragility, but not bone marrow density, and lean muscle mass in overweight individuals with osteoporotic fracture.

## 3. Conclusions

Since its discovery, the myokine irisin has been demonstrated to be involved in the regulation of the muscle-bone-fat axis. More recent evidence indicates that irisin acts on target cells, including bone tissue cells, via αV/β5 integrin receptors. The proliferation, differentiation, and activity of osteoblasts, osteoclasts, and osteocytes are regulated by autocrine and paracrine factors, and potentially by irisin.

Treatment of pre-differentiated osteoblasts with irisin resulted in increased phosphorylation of ERK and p38, which further increased RUNX2, OSX, and ALP/OC protein levels. The activation of these signaling molecules resulted in an increase in osteoblast proliferation, differentiation, and mineralization (Figure 1). In addition, irisin activated the Wnt signaling pathway leading to increased β-catenin protein levels. This enables β-catenin to enter the nucleus and further increase RUNX gene expression, enhancing osteoblast proliferation and differentiation [58,60,62,63,64,66]. In osteoclasts, treatment with irisin inhibited the RANKL receptor thereby leading to decreased osteoclast proliferation and differentiation (Figure 2). Irisin also acts to inhibit NF-κB expression leading to a decrease in osteoclast specific gene expression [58,67]. Irisin treatment had both stimulatory and inhibitory effects in osteocytes. Through binding to αV/β5 receptors, irisin activated the FAK pathway leading to downstream activation of AKT and ERK. These pathways lead to activation of β-catenin, increased ATF4, and reduced sclerostin expression, ultimately resulting in increased bone development. Irisin also had an inhibitory role in osteocytes; irisin inhibited the cleavage of caspases 9 and 3, reducing apoptosis to promote osteocyte proliferation and survival (Figure 3) [42,70,71]. Treatment of chondrocytes with irisin promoted cell growth via increased GAG and inhibited Akt and p38 signaling pathways [72]. In addition, irisin acted to protect chondrocyte survival by repressing autophagy and improving chondrocyte anabolism [73].

In vivo studies have shown increased FNDC5 mRNA and irisin protein levels in bone with exercise. In addition, irisin administration led to increased trabecular and cortical bone volume, thickness, and mass. Correlational animal studies also demonstrated a positive association between increased irisin levels and BMD, and negative association with osteoporosis, OVX, and inflammatory bone disease, highlighting the potential importance of irisin in bone homeostasis and disease management (Figure 4).

Clinical studies point to some dimorphic differences between sexes with females tending to have higher levels of serum irisin in their youth compared to men. However, with menopause, a reduction in serum irisin levels is seen, which is associated with increased risk of osteoporosis and bone fractures. In general, studies have shown reduced serum irisin levels with ageing in both men and women which are associated with increased risk for bone fractures and bone related diseases (Figure 5).

Although, the majority of the in vivo animal studies indicate a role of irisin to increase bone formation [59,69,70,71,73,74,75,76,77,78,79,80,81,82,83,84], (Figure 4) there are a few studies that indicate a negative role of irisin on bone homeostasis [42,67]. Hopefully, with more studies and the use of tissue specific irisin knockout models, the exact role of irisin on bone homeostasis and its potential to be used as a treatment for bone-related conditions will become clearer.

The in vitro and in vivo studies reported here support a role for irisin to increase bone formation however there are some discrepancies. Likewise, the clinical studies presented appear to support the idea that reduced serum irisin levels lead to increased risk of bone fractures and bone diseases in general. Bone diseases, such as osteoporosis and rheumatoid arthritis, can significantly lower an individual’s quality of life and, thus, more research should be performed to examine if exogenous irisin administration can promote bone health in aging populations and individuals with bone-related diseases.

## Figures and Tables

**Figure 1 ijms-22-09136-f001:**
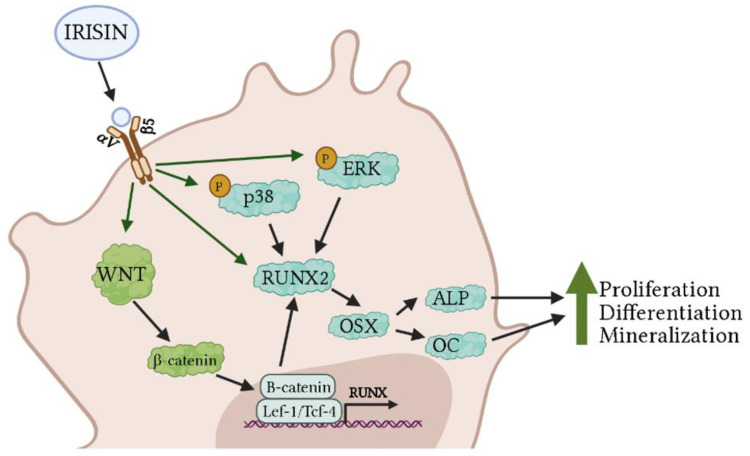
Effects of irisin treatment on osteoblasts in vitro. The figure is based on the data of the studies [58,60,62,63,66] and created using BioRender. Available online: https://biorender.com (accessed February–July 2021).

**Figure 2 ijms-22-09136-f002:**
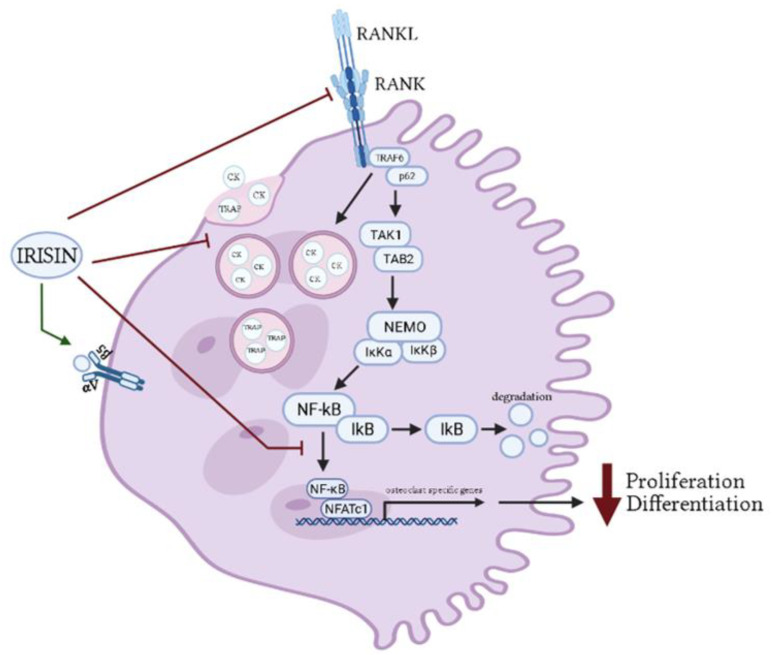
Effects of irisin treatment on osteoclasts in vitro. The figure is based on the data of the studies [58,67,68] and created using BioRender. Available online: https://biorender.com (accessed on 3 February–17 August 2021).

**Figure 3 ijms-22-09136-f003:**
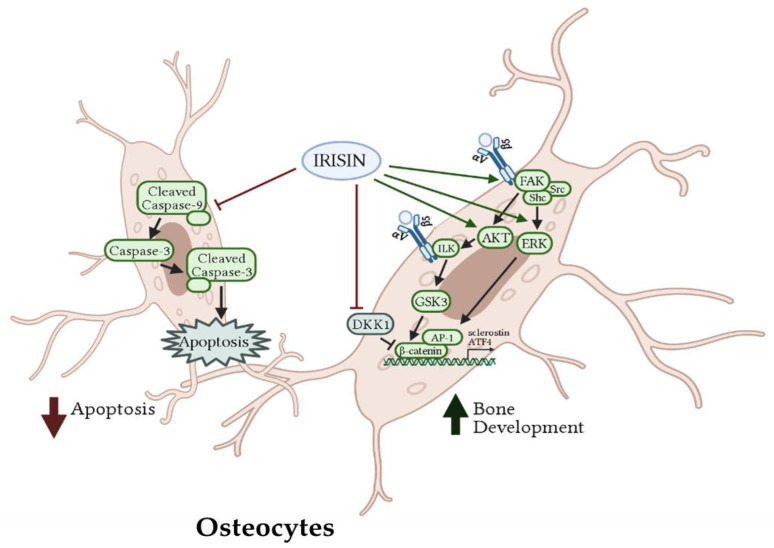
Effects of irisin treatment on osteocytes in vitro. The figure is based on the data of the studies [42,70,71] and created using BioRender. Available online: https://biorender.com (accessed on 3 February–17 August 2021).

**Figure 4 ijms-22-09136-f004:**
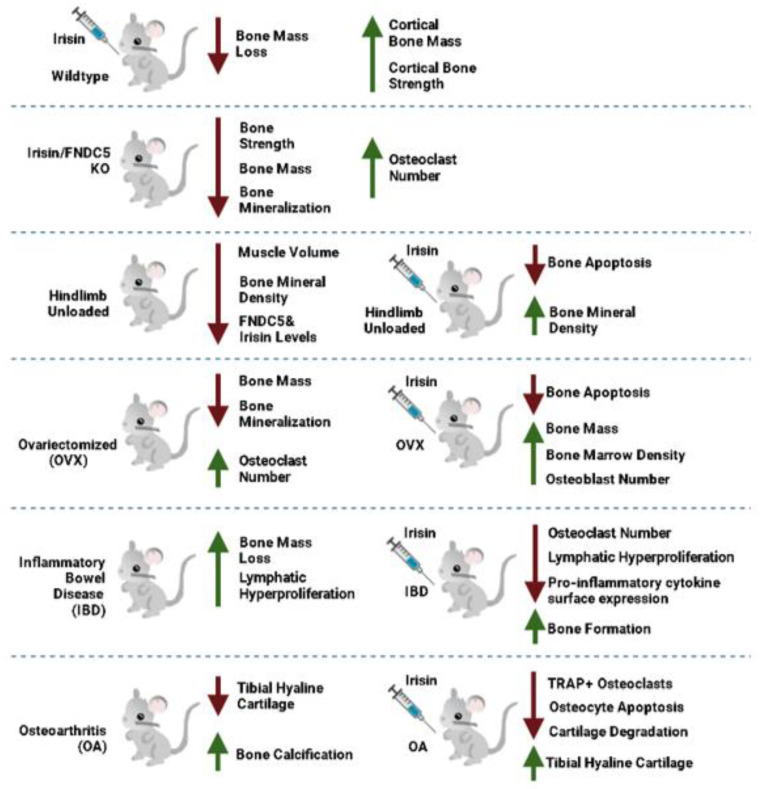
Effects of irisin treatment on bone homeostasis in vivo. The figure is based on the data of the studies [59,69,70,71,74,75,76,77,78,79,80,81,82,83] and created using BioRender. Available online: https://biorender.com (accessed on 3 February–17 August 2021).

**Figure 5 ijms-22-09136-f005:**
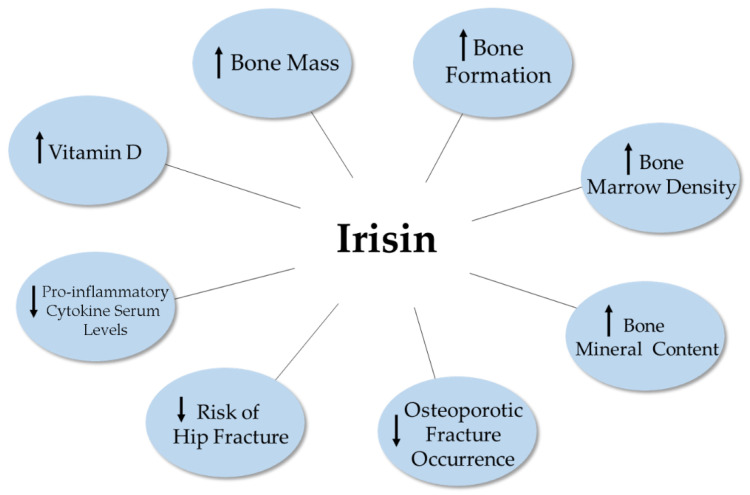
Clinical evidence of the role of irisin on bone homeostasis. Increased serum irisin levels have been associated with increased bone mass, reduced risk of fracture, decreased serum cytokine levels, and increased serum Vitamin D levels. The figure is based on the data of the studies [87,89,90,92,93,94,95,97], and created using BioRender. Available online: https://biorender.com (accessed on 3 February–17 August 2021).

**Table 1 ijms-22-09136-t001:** Role of Irisin on Osteoblasts: in vitro evidence.

Cell Type	Irisin Concentration/Duration	Effect	Reference
Bone marrow stromal cells cultured in exercised conditioned muscle cell-derived media	N/A	↑ Osteoblast differentiation ↑ FNDC5 mRNA and protein ↑ ALP mRNA and protein ↑ Col1a1 mRNA and protein	[56]
Primary rat osteoblasts and MC3T3-E1 cells	Irisin 100 ng/mL; 24 h	↑ Proliferation ↑ Osteoblast differentiation ↑ Runx2 mRNA ↑ Osx mRNA ↑ ALP mRNA ↑ Col1a1 mRNA ↑ OC mRNA ↑ OPG mRNA ↑ p-p38 protein ↑ p-ERK protein	[57]
MC3T3-E1 cells	r-irisin 20 μmol/L; 6 h, 10 days, and 6 weeks	*6 weeks:* ↑ Osteoblast differentiation ↑ Mineralization *10 days:* ↑ Osx mRNA ↑ Runx2 mRNA ↑ SATB2 mRNA ↑ BSP mRNA ↑ Col1a1 mRNA *6 h:* ↑ Nuclear β-catenin ↓ Cytoplasmic β-catenin	[58]
Mouse bone marrow cells primary osteoblasts	Irisin 10, 20, 50, and 100 nM; 24 h	↓ osteoclast formation ↓ RANKL mRNA in primary osteoblasts	[59]
MC3T3E1 cells and bone marrow stromal cells	r-irisin 10 nM; 48 h	↑ Proliferation ↑ Osteoblast differentiation ↑ Osx mRNA ↑ Runx2 mRNA ↑ ALP mRNA ↑ Col1a1 mRNA ↑ ALP-positive colonies ↑ LDHA protein ↑ PDK1 protein ↑ Lactate	[60]
MC3T3E1 cells	r-irisin 20 nM; 14 days	*irisin point mutation:* ↓ R-irisin activity ↓ Proliferation ↓ Osteogenesis	[61]
MC3T3-E1 cells	r-irisin 100 ng/mL; 8 h	↓ Parathyroid hormone receptor mRNA	[62]
Bone marrow mesenchymal stem cells	Irisin 40 μM; 48 h	↑ Osteogenic differentiation ↑ Calcified nodules ↑ Autophagy ↑ Lc3I/II mRNA ↑ Atg5 mRNA ↑ Runx2 mRNA ↑ ALP mRNA ↑ OCN mRNA ↑ β-catenin mRNA ↑ Lef-1 mRNA ↑ Tcf-4 mRNA	[63]
Osteoblasts, osteoclasts, and endothelial cells seeded on Skelite discs	r-irisin 100 ng/mL; 14 days	↑ Osteoblast differentiation ↓ Osteoclastogenesis ↑ Atf4 mRNA ↑ Runx2 mRNA ↑ Osx mRNA ↑ Collagen I mRNA ↑ Osteoprotegerin mRNA	[64]
Pre-osteoblast MC3T3-L1 cells and M0/1 macrophages	Irisin 200 ng/mL; 48 h	↑ M2 macrophage phenotype ↑ Osteogenesis ↑ Mineralized particle formation ↑ CD206 cell surface expression ↑ ARG-1 ↑ TGF-B1 ↓ CD86 cell surface expression ↑ AMPK activity ↑ p-AMPK protein	[65]

**Table 2 ijms-22-09136-t002:** Role of Irisin on Osteoclasts: in vitro evidence.

Cell Type	Irisin Concentration/Duration	Effect	Reference
Pre-osteoclast RAW264.7 cells	r-irisin 20 nmol/L; 3 days	↓ RANKL-induced differentiation ↓ TRAP mRNA ↓ CK mRNA ↓ TRAP-positive multinucleated cells ↓ NFATc1 mRNA and protein ↓ Calcineurin protein ↓ p-Akt protein ↓ p-JNK protein	[58]
Pre-osteoclast RAW264.7 cells	Irisin 2, 5, 10, and 20 ng/mL; 4 h, 24 h, 7 days	↑ Proliferation ↑ Cell size ↑ αV/β5 integrin receptor mRNA ↑ Resorption pit area ↑ Adamts5 mRNA ↑ Lox13 mRNA ↑ Postn mRNA ↑ Igfbp5 mRNA ↑ Tgfb2 mRNA ↑ Sparc mRNA ↑ Atp6vod2 mRNA ↑ c-fps mRNA ↑ Fam102a mRNA ↑ Itgb3 mRNA ↑ Nrf2 mRNA ↑ Dcstamp mRNA ↑ Rank mRNA ↑ Rela mRNA ↑ Rgs12 mRNA ↑ Mst1r mRNA ↑ Itgax mRNA ↓ Slamf8 mRNA ↓ H2-aa mRNA	[67]
Pre-osteoclast Raw264.7 cells and mouse bone marrow monocytes	Irisin 20 nM; 60 min, 4 days	↑ Osteoclast precursor cell proliferation ↓ Osteoclast differentiation ↓ Receptor activators of NF-κB mRNA ↓ NFATc1 mRNA ↓ Cytoplasmic 1 mRNA ↓ CK mRNA ↓ TRAP mRNA ↓ TRAP-positive multinucleated cell number ↓ IkBα protein ↓ p-p65 protein ↓ p-JNK protein ↑ p-p38 protein	[68]
Primary BMSCs isolated from Osx-Cre: F/I KO mice	Irisin 4 and 20 μmol/L; 5 days	↑ Osteoblastogenesis ↑ Mineralization ↓ Osteoclastogenesis ↑ Runx2 mRNA ↑ Satb2 mRNA ↑ Bsp mRNA ↑ Col1a1 mRNA ↑ ALP mRNA ↓ Trap mRNA ↓ MMP9 mRNA ↓ NFATc1 mRNA and protein ↓ p-AKT protein ↓ p-p38 protein ↓ Calcineurin protein ↓ TRAP staining ↓ TRAP-positive multinucleated cell number ↓ Cytosolic β-catenin ↑ Nuclear β-catenin	[69]

**Table 3 ijms-22-09136-t003:** Role of Irisin on Osteocytes: in vitro evidence.

Cell Type	Irisin Concentration/Duration	Effect	Reference
MLO-Y4 cells	Irisin 1–500 ng/mL; 16 h	↓ H_2_O_2_–induced apoptosis ↑ Sclerostin mRNA ↑ p-FAK protein ↑ p-AKT protein ↑ p-CREB protein ↑ p-Zyxin protein	[42]
MLO-Y4 cells	Irisin 100 ng/mL; 24 h	↓ Cyclic stretching-induced apoptosis ↑ Osteocyte proliferation ↑ p-Erk protein ↑ p-p38 protein ↓ Sclerostin mRNA ↑ OPG/RANKL ratio	[70]
MLO-Y4 cells	r-irisin 100 ng/mL; 8 and 24 h	↓ H_2_O_2_–induced apoptosis ↓ Caspase-9 protein ↓ Cleaved caspase-3 protein ↑ Pdpn mRNA ↓ Dkk1 mRNA ↑ Atf4 mRNA ↓ Sclerostin mRNA ↑ Tfam mRNA ↑ Bcl2/Bax ratio ↑ p-ERK protein	[71]

**Table 4 ijms-22-09136-t004:** Role of Irisin on Chondrocytes: in vitro evidence.

Cell Type	Irisin Concentration/Duration	Effect	Reference
hOAC cells	Human r-irisin 25 ng/mL; 7 days	↑ Cell proliferation ↑ GAG production ↑ TIMP-1 mRNA ↑ TIMP-3 mRNA ↓ IL-1 mRNA ↓ IL-6 mRNA ↓ iNOS mRNA ↓ MMP-1 mRNA ↓ MMP-13 mRNA ↑ Type II collagen mRNA and protein ↓ p-p38 protein ↓ p-Akt protein ↓ p-JNK protein ↓ p-NF-κB protein	[72]
Primary mouse chondrocytes isolated from mice with DMM-induced osteoarthritis	Irisin 5 and 10 ng/mL; 6 and 72 h	↑ Proliferation ↑ Mitophagy ↑ Autophagosome formation ↓ ROS production ↑ FNDC5 expression ↑ LC3 expression ↑ PCNA immunoreactivity ↑ GAG production ↑ ECM accumulation ↑ Collagen II mRNA ↑ Aggrecan mRNA ↑ Sox9 mRNA ↓ MMP9 mRNA ↓ CEGF mRNA ↑ PGC-1a mRNA ↑ Tfam mRNA ↑ ATP production ↑ Mito membrane potential ↑ Sirt3 protein ↑ UCP-1 protein	[73]

**Table 5 ijms-22-09136-t005:** Role of Irisin on bone homeostasis: in vivo evidence.

Animal Model	Irisin Concentration/Duration	Effect	Reference
C57BL6 male mice	Hind-limb suspension + irisin injections (100 µg/kg b.w.); once a week for 4 weeks	↓ Disuse-induced bone mass loss ↑ BV/TV ratio ↑ Femoral and tibia radiodensity ↑ Polar moment of inertia ↑ Bending strength	[74]
C57BL/6J mice and FNDC5-KO mice	r-irisin injection (1 mg/kg b.w.); 6 days	↑ Bone and plasma sclerostin mRNA ↑ Ucp1 mRNA and protein *FNDC5-KO:* ↓ RANKL mRNA ↑ Femoral bone mass ↑ Femoral connectivity	[42]
C57BL/6J mice with forced expression of FNDC5	N/A	↓ Cortical bone area ↓ Trabecular bone volume ↓ Cortical thickness ↑ Osteoclast number and size	[67]
Mice lacking functional irisin	N/A	↓ Bone strength ↓ Bone mass ↑ Osteoclast number ↑ RANKL surface expression ↓ Browning response ↑ Intraperitoneal white adipose cell size ↑ Hyperlipidemia ↑ Insulin resistance ↑ LDL-cholesterol level ↓ HDL-cholesterol level	[75]
Osx-Cre:FNDC5/irisin KO mice	r-irisin (undisclosed dosage); 14 days	↓ Irisin mRNA and protein ↓ BMD ↓ Bone development ↓ Bone mineralization ↓ Trabecular bone mass ↓ Trabecular bone area ↓ Osteoblast number ↓ Cortical BMD ↓ Trabecular BV/TV ↑ Cortical BS/BV ↓ Runx2 mRNA ↓ Bsp mRNA ↓ Osx mRNA ↓ Alp mRNA ↑ Cathepsin K mRNA ↑ Mmp9 mRNA ↑ Trap mRNA *r-irisin:* ↑ Bone strength ↑ Osteoblastogenesis ↓ Osteoclastogenesis	[69]
Wild-type C57BL/6J	2 weeks voluntary Wheel-running exercise Irisin injection 3.24 ng daily	↑ FNDC5 mRNA in bone tissue ↑ Irisin	[58]
Hindlimb unloaded and sciatic neurectomic mice	N/A	↓ Trabecular BMD ↓ Muscle volume ↓ Soleus FNDC5 mRNA Soleus FNDC5 = ↑ Tibia trabecular BMD Soleus FNDC5 = ↓ RANKL mRNA	[59]
Hindlimb unloaded mice and primary osteoblasts with stimulated microgravity	Primary osteoblasts: r-irisin (1 nM); 14 days	*Hindlimb unloaded mice:* ↑ Femur bone loss ↓ Mineral apposition rates ↓ Alp mRNA ↓ ColIa1 mRNA ↓ Fndc5 mRNA *Primary osteoblasts:* ↑ Osteoblast differentiation ↑ Osteoblast proliferation ↑ Alp mRNA ↑ ColIa1 mRNA ↑ ALP activity ↑ Mineralization ↑ CyclinA2, D1, and E1 mRNA ↑ CDK2 and 12 mRNA ↑ β-catenin mRNA	[76]
Hindlimb unloaded Sprague–Dawley rats	r-irisin (18 ng/mL); three times per week for 8 weeks	↑ Bone homeostasis ↑ Bone formation rate ↓ Osteoclast surface ↓ TNF-α level ↓ IL-17 level ↓ RANKL level ↓ Sclerostin level	[77]
Hindlimb unloaded C57BL6 male mice	hindlimb suspension irisin injections (100 µg/kg b.w.); 4 weeks	↓ Osteocyte apoptosis ↑ Bcl2/Bax ratio ↓ Caspase 3 protein ↓ Caspase 9 protein	[71]
C57BL/6J mice and FNDC5-KO mice	r-irisin injection (1 mg/kg b.w.); 6 days cRGDyK (1 mg/kg)	*r-irisin:* ↑ Bone and plasma sclerostin mRNA *Co-injection r-irisin and cRGDyK:* ↓ Ucp1 mRNA and protein ↓ Dio2 mRNA	[42]
Ovariectomized (OVX) mice	r-irisin (100 ug/kg b.w.); 5 weeks	↓ Trabecular bone loss ↑ Greater bone microarchitecture ↑ BMD ↑ BV/VR ↑ Connection density ↑ Bone stiffness ↑ Osteoblast number ↓ Osteoclast number ↑ Osteocalcin serum level ↓ TRAP serum level	[78]
Postmenopausal Sprague–Dawley rats with osteoporosis	Irisin (1 mmol/L)	↑ Trabecular thickness ↑ Trabecular number ↑ Trabecular BMD ↓ Osteoblast apoptosis ↓ ALP serum level ↓ Caspase-2 mRNA and protein ↓ NLRP3 mRNA and protein ↑ RUNX2 mRNA ↑ OC mRNA ↑ Bcl-2 mRNA and protein ↑ Nrf2 mRNA and protein	[79]
OVX mice that underwent moderate intensity treadmill exercise	N/A	↑ Irisin protein ↑ Fndc5 mRNA Fndc5 mRNA = ↑ Trabecular BMD	[80]
Androgen deficient and osteopenic mice	r-irisin (100 ug/kg b.w.); once a week for 8 weeks	↑ Trabecular BMD Soleus Fndc5 mRNA = ↑ Trabecular BMD Soleus Fndc5 mRNA ≠ ↑ Cortical BMD	[81]
Sprague–Dawley rats with IBD	r-irisin (18 ng/mL b.w.); 2 times per week for 3 weeks	↓ Gut inflammation ↓ Bone inflammation ↑ Bone structure ↑ Bone formation rate ↓ Lymphatic hyperproliferation ↓ Osteoblast surface expression ↓ TNF-α protein ↓ RANKL protein	[82]
Sprague–Dawley rats with IBD	r-irisin (18 ng/mL b.w.); 2 times per week for 3 weeks	↓ DSS-stimulated colon inflammation ↑ Bone formation ↓ Osteoclast surface expression ↓ TNF-α positive expression ↓ RANKL positive expression ↓ OPG positive expression ↓ IL-6 positive expression	[83]
C57BL6 mice	r-irisin (100 µg/kg b.w.); 4 weeks	↓ Cartilage degradation ↑ Tibial hyaline cartilage ↓ Calcified cartilage ↑ BV/TV ↑ Trabecular bone number ↑ Connective density ↓ Osteocyte apoptosis ↓ TRAP positive expression ↓ MMP-13 ↓ Caspase 3	[70]
C57BL6 mice with destabilized medial meniscus	r-irisin (10 µL) injected into injured knee joint	↓ Cartilage injury ↓ Synovitis ↑ Walking ability ↑ LC3 expression ↓ Proliferating cell nuclear antigen ↓ TUNEL	[73]
Sprague–Dawley rats fed an HFD and underwent 8 weeks exercise regimen	N/A	↑ Body weight ↓ BMD ↓ Femur and tibia microstructure *Exercise:* ↑ BMD ↑ Femur and tibia microstructure ↑ β-catenin ↑ PGC-1α ↑ FNDC5 ↑ Osteocalcin serum level ↑ Irisin serum level	[84]

**Table 6 ijms-22-09136-t006:** Role of Irisin on bone homeostasis: clinical evidence.

Clinical Model	Effect	Reference
Healthy children	Irisin = ↑ bone mineral content Irisin = ↑ osteocalcin serum level Irisin = ↑ CTX serum level Irisin = ↓ DKK-1 serum level	[87]
Adolescent and adult swimmers	Both groups: ↑ IL-1β ↑ IL-10 Adults: ↑ IL-6 ↑ TNF-α ↓ Irisin	[88]
Athletic Caucasian football players	Irisin = ↑ BMD Irisin = ↑ bone mineral content	[89]
Women that are postmenopausal and with low bone mass	↓ Irisin serum level Age = ↓ irisin Parathyroid hormone = ↓ irisin Creatinine = ↓ irisin Osteoporotic fracture occurrence = ↓ irisin	[90]
Women with postmenopausal osteoporosis	↓ Irisin serum level ↓ Chemerin serum level ↑ C-reactive protein level	[91]
Chinese elderly population with extremely high hip bone marrow density	↑ Irisin serum levels (males) Irisin = ↑ BMD Irisin = ↑ triglyceride level	[92]
Geriatric Chinese men with osteoporosis or osteopenia	↓ Irisin serum level Irisin = ↑ BMD	[93]
Patients with RA	↓ Irisin serum level ↑ Low-fracture bone fractures ↑ Extra-articular manifestations ↑ Function joint failure ↓ Vitamin D serum level	[94]
Patients with end-stage osteoarthritis	↑ Cartilage damage ↑ 8-OHdG expression ↑ TUNEL expression ↓ LC3-II expression ↓ FNDC5 expression	[73]
Elderly Chinese women that had suffered minimal trauma hip fractures	↓ Irisin serum level Irisin = ↑ BMD Low irisin level = ↑ risk of hip fracture	[95]
Individuals with total hip/knee replacement and osteopenia/osteoporosis	Age = ↓ irisin serum level Irisin = ↑ osteocalcin mRNA FNDC5 = ↑ total femur BMD FNDC5 = ↑ femoral neck BMD	[96]
Individuals with osteoporosis and osteopenia	Lumbar bone marrow density = ↑ serum irisin level Lumbar T-score = ↓ serum irisin level	[97]
Overweight individuals with a previous vertebral osteoporotic fracture	↓ Irisin serum level Irisin ≠ BMD Irisin ≠ daily physical activity	[98]

## Data Availability

Not Applicable.

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
