# Peer review of "Role of the Myokine Irisin on Bone Homeostasis: Review of the Current Evidence"

_ijms, 2021, doi:10.3390/ijms22179136_

Round 1

Reviewer 1 Report

  The review “Role of the myokine irisin on bone homeostasis: review of the current evidence” is well-written, and provides a new insight into the effects of irisin on bone homeostasis.  I assess that the submitted manuscript satisfies the academic level of International Journal of Molecular Sciences for being published in the journal.  

This is an "excellent" review I have no hesitation in recommending it for publication.

Author Response

We thank the reviewer for the assessment of our manuscript.  We are pleased that this reviewer recommended our manuscript for publication.  The revised manuscript is further improved.

Reviewer 2 Report

The authors carry out an exhaustive narrative review of the role of a myokin, irisin, on bone homeostasis. The analysis is carried out at the cellular level (osteoblasts, osteoclasts, osteocytes and chondrocytes) with a didactic representation on each of these cells. In vivo studies in animals and humans are also discussed. The discussion summarizes the main findings.

The authors carry out an exhaustive narrative review on the role of a myokin, irisin, in bone metabolism. It is a recently described myokin whose influence on bone remodeling has not been sufficiently clarified. The approach is carried out evaluating the effect of myokin on three levels: cellular, experimental in vivo and in humans. This approach is interesting although the effect in humans is less uniform. The bibliography reviewed is extensive, reflecting in tables the most outstanding aspects of each of the articles. This is an interesting element but it can cause a loss in the integrated knowledge of these articles.

The cellular approach is the simplest and it has also been accompanied by a graphic representation of the effects on each of these cells. Although it simplifies the effect, it is very interesting for non-experts in this myokine. The experimental results in animals and humans are less conclusive and the data more difficult to interpret. However, if a graphical representation of its effects could be made it could help better compression. In the discussion, more emphasis should probably be placed on the clinical consequences of this myokine and its possible therapeutic effects.

Author Response

We thank the reviewer for their comments. We have added a figure (Figure 4) shown below, into the review that highlights the effects of irisin on bone homeostasis based on animals studies.

Figure 4: Effects of irisin treatment on bone homeostasis in vivo. The figure is based on the data of the studies [59,70-72,75-84] and created using BioRender.com.

In the discussion, more emphasis should probably be placed on the clinical consequences of this myokine and its possible therapeutic effects.

We thank the reviewer for the suggestion. We have created and added figure 5. We also have added more emphasis on the clinical consequences and therapeutic effects of irisin in the discussion.

Please see below and in the revised manuscript pages 24 and 26; lines 746-752.

Figure 5: Clinical evidence of the role of irisin on bone homeostasis. Increased serum irisin levels have been associated with increased bone mass, reduced risk of fracture, decreased serum cytokine levels and increased serum Vitamin D levels. The figure is based on the data of the studies [88,90,91,93-96,98], and created using BioRender.com.

“The in vitro and in vivo studies reported here support a role for irisin to increase bone formation however there are some discrepancies. Likewise, the clinical studies presented appear to support the idea that reduced serum irisin levels lead to increased risk of bone fractures and bone diseases in general. Bone diseases such as osteoporosis and rheumatoid arthritis can significantly lower an individual’s quality of life and thus more research should be performed to examine if exogenous irisin administration can promote bone health in aging populations and individuals with bone related diseases.”
